# Growth Promotion of *Phaseolus vulgaris* and *Arabidopsis* *thaliana* Seedlings by Streptomycetes Volatile Compounds

**DOI:** 10.3390/plants11070875

**Published:** 2022-03-25

**Authors:** Daniel Alonso Pérez-Corral, José de Jesús Ornelas-Paz, Guadalupe Isela Olivas, Carlos Horacio Acosta-Muñiz, Miguel Ángel Salas-Marina, David Ignacio Berlanga-Reyes, David Roberto Sepulveda, Yericka Mares-Ponce de León, Claudio Rios-Velasco

**Affiliations:** 1Centro de Investigación en Alimentación y Desarrollo A.C., Unidad Cuauhtémoc, Av. Río Conchos, S/N, Parque Industrial, Cd. Cuauhtémoc C.P. 31570, Chihuahua, Mexico; danielpc18@gmail.com (D.A.P.-C.); jornelas@ciad.mx (J.d.J.O.-P.); golivas@ciad.mx (G.I.O.); cacosta@ciad.mx (C.H.A.-M.); dberlanga@ciad.mx (D.I.B.-R.); dsepulveda@ciad.mx (D.R.S.); yerickamapdl@gmail.com (Y.M.-P.d.L.); 2División de Ingeniería, Universidad de Ciencias y Artes de Chiapas, Carretera Villacorzo-Ejido Monterrey Km 3.0., Tuxtla Gutiérrez C.P. 30520, Chiapas, Mexico; miguel.salas@unicach.mx

**Keywords:** beneficial bacteria, sustainable agriculture, bacterial metabolites, volatile organic compounds (VOCs), plant–*Streptomyces* interactions, bean, plant hormones

## Abstract

*Streptomyces* are recognized as antipathogenic agents and plant-growth-promoting rhizobacteria. The objective of this study was to evaluate the capacities of four antifungal *Streptomyces* strains to: produce the substances that are involved in plant growth; solubilize phosphates; and fix nitrogen. The effects of the volatile organic compounds (VOCs) that are emitted by these strains on the growth promotion of *Arabidopsis thaliana* and *Phaseolus vulgaris* L. (var. Pinto Saltillo) seedlings were also tested. All of the *Streptomyces* strains produced indole-3-acetic acid (IAA) (10.0 mg/L to 77.5 mg/L) and solubilized phosphates, but they did not fix nitrogen. In vitro assays showed that the VOCs from *Streptomyces* increased the shoot fresh weights (89–399%) and the root fresh weights (94–300%) in *A*. *thaliana* seedlings; however, these effects were less evident in *P. vulgaris.* In situ experiments showed that all the *Streptomyces* strains increased the shoot fresh weight (11.64–43.92%), the shoot length (11.39–29.01%), the root fresh weight (80.11–140.90%), the root length (40.06–59.01%), the hypocotyl diameter (up to 6.35%), and the chlorophyll content (up to 10.0%) in *P. vulgaris* seedlings. 3-Methyl-2-butanol had the highest effect among the ten pure VOCs on the growth promotion of *A. thaliana* seedlings. The tested *Streptomyces* strains favored biomass accumulation in *A. thaliana* and *P. vulgaris* seedlings.

## 1. Introduction

The current agronomic practices involve the extensive use of mineral fertilizers, which are usually expensive, and which exert negative impacts on the environment [1]. Fortunately, the interest in environmentally friendly and sustainable agricultural practices is increasing, and the use of beneficial microorganisms with fertilizing activities is becoming popular among farmers [2]. These microorganisms are collectively known as “plant-growth-promoting rhizobacteria” (PGPR). They can be applied to seeds, plants, and soil to colonize the rhizosphere, and they cause many beneficial effects in crops [3,4,5]. They favor plant growth and crop yields by direct mechanisms, including the production of phytohormones, which regulate the levels of the plant hormones [6], bioremediate the soil [7], and increase the mineral content in the plants by producing siderophores and by favoring nitrogen fixation and phosphorus solubilization [1].

These microorganisms also exert several beneficial effects in plants by indirect mechanisms, which include the production of anti-phytopathogenic compounds, the biosynthesis of enzymes with antiparasitic activity, and the activation of the defense systems of the plants, and especially the induction of systemic resistance [1,8]. The *Streptomyces* genus is the most widely distributed in nature, and it is studied in agriculture and in other sectors [9]. Several studies have demonstrated that the bacteria of this genus are able to produce nonvolatile compounds that promote plant growth (e.g., indole-3-acetic acid (IAA), siderophores, and gibberellic acid) [10,11], and that they inhibit the growth of phytopathogens (e.g., tubercidin, valinomycin, polyenes, and reveromycins A and B, among others), and especially fungi [12,13]. The volatile compounds (VOCs) that are emitted by *Streptomyces* also contribute to these effects, although they have received little attention in this regard. Actinobacteria from the *Streptomyces* genus produce a wide diversity of VOCs, including alkanes, alkenes, aromatic hydrocarbons, alcohols, sulfides, ketones, esters, and terpenes [13,14,15]. The antipathogenic effects of VOCs from *Streptomyces* have mainly been attributed to *trans*-2-hexenal and dimethyl disulfide [13,16,17,18]. However, the specific VOCs from the *Streptomyces* that are responsible for the plant-growth-promoting effects remain unknown [19,20]. Therefore, the aim of the study was to evaluate the capacities of four antifungal *Streptomyces* strains to produce nonvolatile and VOCs that are involved in the promotion of plant growth, and to evaluate the effects of the selected VOCs from these *Streptomyces* strains on the growth of *Arabidopsis thaliana* and *Phaseolus vulgaris* seedlings.

## 2. Results

### 2.1. Phytohormone Production and Solubilization of P and Fixation of N by Streptomyces Strains

The four *Streptomyces* strains produced indole-3-acetic acid (IAA) (Table 1), although the quantity of this compound varied with the strain. *Streptomyces cangkringensis* (CIAD–CA07 strain) produced the highest quantity (77.5 mg/L) of IAA. The results of the nitrogen fixing and the phosphate solubilization by the *Streptomyces* strains, as well as the detection of the catechol and hydroxamate-type siderophore, are shown in Table 1.

### 2.2. In Vitro Plant-Growth-Promoting Activity of Streptomyces VOCs in A. thaliana and P. vulgaris Seedlings

The VOCs of the tested *Streptomyces* strains increased the biomasses of both plant species. The appearance of *A. thaliana* seedlings after 10 days of exposure to *Streptomyces* VOCs is shown in Figure 1. The VOCs significantly increased the root fresh weight (Figure 2). The shoot fresh weight was also significantly increased by the VOCs (Figure 2). The VOCs of *S. misionensis* (CIAD-CA27 strain) caused the highest increase in the shoot fresh weight (399%) and the root fresh weight (300%), compared to the control *A. thaliana* seedlings. The number of leaves and the length of the primary root in the *A. thaliana* seedlings were not influenced by the *Streptomyces* VOCs.

The appearance of a bean seedling after 5 days of exposure to *Streptomyces* VOCs is shown in Figure 3. The VOCs of the *Streptomyces* strains showed lower plant-growth-promoting activity in the bean seedlings as compared to *A*. *thaliana.* The VOCs from *S. kanamyceticus* (CIAD-CA45 strain) caused significant increases in the root fresh weight (44.73%) and the primary root length (37.85%) of the bean seedlings (Figure 4). On the other hand, the VOCs of *S**. misionensis* (CIAD-CA27 strain) only increased (21.18%) the primary root of the *P. vulgaris* seedlings (Figure 4).

### 2.3. In Situ Effect of Streptomyces on Growth of Bean Seedlings

The tested *Streptomyces* strains significantly improved the in situ growth of the bean seedlings, especially the root length and root fresh weight (Figure 5). *S.*
*kanamyceticus* CIAD–CA45 and *S.*
*cangkringensis* CIAD–CA07 caused the highest increases in the shoot fresh weight (31.07–43.92%) (Figure 5). No differences were observed in the hypocotyl diameter and the chlorophyll content in the bean seedlings.

### 2.4. In Vitro Growth Promotion of A. thaliana Seedlings by Pure VOCs

The appearance of *A*. *thaliana* seedlings after 10 days of exposure to several concentrations of the pure VOCs that were previously identified in the tested *Streptomyces* strains is shown in Figure 6. The 3-methyl-2-butanol caused the highest increase in the shoot fresh weight (168%) and the root fresh weight (224%) at the highest concentration (50 μL/L). The lowest concentration (5 μL/L) of dimethyl disulfide (DMDS) also increased these variables, but this effect was lower (53% and 93%, respectively) than that of 3-methyl-2-butanol (Figure 7). In contrast, *trans*-2-hexenal negatively affected the growth of the *A. thaliana* seedlings (Figure 6 and Figure 7). The other VOCs did not influence the plant growth.

## 3. Discussion

The study evidenced the capacities of four *Streptomyces* strains to produce compounds (IAA, siderophores, and VOCs) that are associated with the promotion of plant growth. Additionally, these strains favored the solubilizing of phosphates. These effects improved the growth of *A. thaliana* and *P. vulgaris* seedlings.

The tested *Streptomyces* strains favored the solubilization of phosphates. This effect might cause increases in the quantity of the assimilable forms of this mineral in soil, where insoluble forms of P generally prevail [21,22]. These strains also might increase the efficiency of the direct fertilization of crops with P-based chemical products, which only allow plants to use 30% of the applied P [21,23]. Strains from other genii and from other *Streptomyces* strains can solubilize phosphates [24,25]. These microorganisms release the P from insoluble forms by different mechanisms, including by the production of compounds that are able to dissolve phosphates (e.g., organic acids, hydroxyl ions, CO^2^), which releases enzymes that metabolize the insoluble forms of P into soluble forms, and which degrade the plant substrate [1,26].

The tested strains did not fix N. Dahal et al. [27] reported nitrogen-fixing *Streptomyces* strains for the first time, and they observed that only 11% of them had such an ability, which was attributed to the presence of the *nifH* gene in these strains. This gene and other genes, such as *ni*fD and *ni*fK, encode multiple subunits of the nitrogenase enzyme [28]. The expression or overexpression of the nitrogenase enzyme complex causes the transformation of the dinitrogen gas into ammonium equivalents, which are needed for the biosynthesis of the essential cellular macromolecules in plants [1,22,27,28]. N fixing by these kinds of microorganisms causes increases in the protein content in the roots and shoots, as well as higher levels of chlorophyll b and carbohydrates in plants [23].

The tested *Streptomyces* strains produced IAA. Overall, *S*. *misionensis* CIAD–CA07 produced the highest quantity of this compound (77.5 mg/L), which was in the range that is generally reported for other *Streptomyces* strains (between 56.51 and 136.5 mg/L) [26]. However, there are some *Streptomyces* strains that produce very low quantities of IAA, such as *S*. *roseocinereus* MS1B15 (6.34 mg/L) [24]. In our study, *S*. *kanamyceticus* CIAD–CA48 also produced a very low quantity of IAA (10 mg/L). The production of phytohormones, including IAA, by *Streptomyces* strains has been widely reported [29]. The production of IAA by rhizospheric microorganisms, including *Streptomyces* strains, depends on the availability of root exudates, which are rich in the nutrients that are required for these microorganisms [30]. The positive effects of IAA in plants have been extensively documented. Each plant species responds differently to high or low concentrations of IAA, although generally, the best stimulatory effect is achieved with low concentrations of this compound [25,31]. Thus, IAA-producing microorganisms, including *Streptomyces,* have been proposed as biofertilizers for the large-scale exploitation of crops [32].

PGPRs produce VOCs that stimulate plant growth by different mechanisms, including by increasing the photosynthetic capacities of plants, by regulating the biosynthesis of phytohormones, and by inducing systemic disease resistance [20]. In our study, the tested *Streptomyces* strains significantly promoted the in vitro growth of *A*. *thaliana* seedlings, especially the accumulation of the biomass in the roots (root fresh weight: 94–300%) and aerial part (89–399%). Similarly, Cordovez et al. [33] report increases in the shoot fresh weight (230%) and the root fresh weight (100%) of *A*. *thaliana* seedlings that were exposed to the VOCs of *S*. *lividans* 1326, with these increases being attributed to the VOC acetoin (3-hydroxy-2-butanone). Acetoin is among the first VOCs that were considered as plant growth promoters [34]. The VOCs of other bacterial species also favored the in vitro growth of *A*. *thaliana*. Pérez-Flores et al. [35] report increases in the root fresh weight (114.28%), the shoot fresh weight (70%), and the number of leaves (50%) in *A*. *thaliana* seedlings exposed to VOCs from a *Bacillus methylotrophicus* strain, and these increases were attributed to an increase in the auxins in the apical meristems and leaf veins, which suggests that the biosynthesis and transport of IAA increased in the plants. On the other hand, Lee et al. [36] exposed *A. thaliana* seedlings to the VOCs of 20 *Trichoderma* strains and observed that the VOCs that were emitted by *Trichoderma aggressivum* (IMI 393,970 strain) caused the highest increase (37.1%) in the shoot fresh weight.

The VOCs of the tested *Streptomyces* strains also improved the in vitro growth of *P*. *vulgaris* seedlings. However, the effect was less intense compared with the outcome that was observed with *A. thaliana* (Figure 3 and Figure 4), which demonstrates that the actions of these VOCs strongly depend on the plant species [35]. To date, the biological functions of microbial VOCs, including those emitted by *Streptomyces,* are not fully understood; however, it has been suggested that VOCs might be a medium for inter- and intra-organism communication [37]. This might explain the differences in the plant growth that was observed for *P*. *vulgaris* and *A. thaliana*. Several roles for microbial VOCs have recently been recognized, including their action on plant growth [35]. The low molecular weight and high vapor pressure of microbial VOCs at normal temperatures are attributes that favor their evaporation and diffusion in different substrates and environments, and that favor their interactions with microorganisms and plants [38].

Actinomycetes are considered to be broad-spectrum PGPR [39,40]. In our in situ experiment, significant increases in the shoot fresh weight (11.64–43.92%), the shoot length (11.39–29.01%), the root fresh weight (80.11–140.90%), the root length (40.06–59.01%), the hypocotyl diameter (up to 6.35%), and the chlorophyll content (up to 10.0%) were observed in bean seedlings. AlAli et al. [41] observed an increase of 55% in the primary root length of white bean plants (*P*. *vulgaris*) that were inoculated with *Streptomyces* sp. (strain FCL6), and they attributed this positive effect to the production of IAA and gibberellin by *Streptomyces.* Kumari et al. [42] observed increases in the shoot length (32.26–13.38%) and main root length (84.60–61.94%) of mung bean plants (*Vigna radiata* (L.) R. Wilczek) that were inoculated with *Pseudomonas aeruginosa* (BHU B13–398 strain) or *Bacillus subtilis* (BHU M strain). They also report an increase in the chlorophyll content, and they infer a possible increase in the photosynthetic efficiency of the plant [42]. The positive influence of the *Streptomyces* strains, especially in the increases in the growth parameters of *P. vulgaris,* might be a consequence of their VOCs (Figure 5) and of their capacity to solubilize phosphates (Table 1). On the other hand, the positive effects of IAA and siderophores on the plant morphology and development have been widely demonstrated, especially in the increase in the root biomass [43]. Thus, in our study, the promotion of the morphological and yield parameters on bean plants by *Streptomyces* strains possibly occurred by direct mechanisms, especially those driven by IAA and siderophore production, or by altering the hormone balance in the plant [42]. The *Streptomyces* strains probably survived in the bean rhizosphere and enhanced the soil health [44].

Only the pure VOCs, 3-methyl-2-butanol and DMDS, promoted the in vitro growth of *A. thaliana*. DMDS is produced by several microorganisms, including *Streptomyces,* and it is considered as a plant growth promoter at low concentrations. This VOC induces changes in the auxin levels, which presumably increases the expression of the early response genes that are associated with the signaling of this phytohormone [45]. Plyuta et al. [46] exposed *A. thaliana* seedlings to DMDS at concentrations of 25 and 50 μM and observed growth suppression. In our study, positive results were observed with *A. thaliana* at the lowest concentration (5 μL/L) of DMDS; however, the highest concentration (50 μL/L) of this compound did not influence the plant growth. 3-Methyl-2-butanol at the highest concentration (50 μL/L) significantly influenced the growth on *A. thaliana*. Gamboa-Becerra et al. [47] evaluated the plant-growth-promoting effect of a similar VOC (3-Methyl-1-butanol) at concentrations of 25, 50, and 100 μM in *A. thaliana*, and they observed increases in the shoot fresh weight and the root fresh weight, especially at a concentration of 50 μM. The authors of [47] also observed the increased activity of GUS in treated plants. In our study, the growth promotion that was caused by the tested pure volatiles in *A. thaliana* was lower than that documented in other studies with the *Streptomyces* strains. This suggests that, besides the positive effects of individual VOCs, the synergistic effects of several VOCs, and the effects of nonvolatile compounds, are involved in the plant-growth-promoting effects of *Streptomyces*.

## 4. Material and Methods

### 4.1. Reagents and Microorganism Strains and Seeds

All reagents (analytical grade) were purchased from Sigma Aldrich Corp. (St. Louis, MO, USA). The four *Streptomyces* strains (*S. cangkringensis* CIAD–CA07; *S. misionensis* CIAD–CA27; *S. kanamyceticus* CIAD–CA45; and *S. kanamyceticus* CIAD–CA48) that were used in the study were previously isolated from the rhizosphere of apple trees (*Malus domestica* Borkh.) and bean (*Phaseolus vulgaris*) and maize (*Zea mays* L.) plants in Chihuahua, Mexico, and were characterized for their morphological, molecular and biochemical characteristics, according to [48]. The strains are deposited in GenBank under the accession numbers: MK968576.1; MK968589.1; MK968601.1; and MK968603.1, respectively. These strains were selected for their in vitro anti-phytopathogenic activity [13,48].

Bean seeds (*P. vulgaris* var. Pinto Saltillo) were obtained from the Centro de Investigación en Alimentación y Desarrollo A. C. (CIAD) Campus Cuauhtémoc. The seeds of *Arabidopsis thaliana* (wild-type genotype Col-0) were provided by the Centro de Investigación y de Estudios Avanzados (CINVESTAV) Instituto Politécnico Nacional (IPN), Irapuato, Mexico.

### 4.2. Evaluation of the Nitrogen Fixing and Phosphate Solubilization Activities of Streptomyces Strains

The nitrogen-fixing activity of the *Streptomyces* strains was determined qualitatively on a glucose nitrogen-free mineral medium containing bromothymol blue. This medium was prepared with 10 g of glucose; 1 g of K_2_HPO_4_; 0.2 g of MgSO_4_; 0.2 g of NaCl; 1 g of CaCO_3_; 0.005 g of NaMoO_4_; 0.1 g of FeSO_4_; 20 g of agar; and 1 L of sterile distilled water (SDW) (Osmosis and UV treatment (Ultrapure Water Production System (FESTA Model HGL-UP-TOC-100, Chihuahua, Mexico)). The pH of the medium was adjusted to 7. Circle explants (7 mm in diameter) of each *Streptomyces* strain were inoculated onto Petri dishes (90 mm in diameter) that contained the medium described above and were incubated at 28 ± 1 °C for 7 d. Color changes in the colonies indicated strains with nitrogen-fixing activity [49]. The phosphate solubilization activity of the *Streptomyces* strains was tested by plate assays by using a phosphate growth medium that consisted of 10 g of glucose; 5 g of Ca_3_(PO_4_)_2_; 5 g of MgCl_2_·6H_2_O; 0.25 g of MgSO_4_·7H_2_O; 0.2 g of KCl; 0.1 g of (NH_4_)_2_SO_4_; and 1 L of SDW. The pH of the medium was adjusted to 7. Circle explants (7 mm in diameter) of each *Streptomyces* strain were inoculated onto the plates that contained the specific medium and were incubated at 28 ± 1 °C for 7 d. The halo formation around the *Streptomyces* colonies after this incubation period was considered as indicative of phosphate solubilization activity [50].

### 4.3. Indole-3-Acetic Acid (IAA) and Siderophore Production

The IAA production by the *Streptomyces* strains was determined by using the methods that were described previously in [30,44]. Fifty microliters (~2 × 10^6^ spores mL^−1^) of a spore suspension of each *Streptomyces* strain were inoculated in 15 mL conical tubes containing 4 mL of ISP2 broth supplemented with L-tryptophan [51]. Then, they were incubated at 28 ± 1 °C with constant shaking at 150 rpm for 7 d. The cultures were centrifuged (10,000× *g* for 15 min), and 1 mL of the supernatant was placed in a glass test tube to be immediately mixed with 2 mL of Salkowski’s reagent (1 mL of 0.5 M iron trichloride, and 50 mL of 35% perchloric acid). This mixture was maintained in darkness for 30 min at room temperature (approximately 23 ± 2 °C). The development of a pink color was considered as a positive indicator for IAA production. The color intensity was determined by UV–Vis spectrophotometry (Evolution™ 300 UV-Vis spectrophotometer, Thermo Scientific, Waltham, MA, USA) at 535 nm, and the IAA concentration was determined by using a calibration curve that was constructed with several dilutions of pure IAA [52]. The ability of the *Streptomyces* strains to produce siderophores was tested on chrome azurol S agar [53]. Circle explants (7 mm in diameter) of each *Streptomyces* strain were inoculated on plates that contained the specific medium, and were subsequently incubated at 28 ± 1 °C for 7 d. After incubation, the development of orange halos (color change from blue to orange) around the colonies indicated the production of siderophores (positive reaction). The detection of catechol and hydroxamate-type siderophore was performed according to Lee et al. [54].

### 4.4. In Vitro Growth Promotion of Arabidopsis thaliana by Streptomyces VOCs

The effect of the VOCs that are emitted by the *Streptomyces* strains on the plant growth promotion was tested in vitro in *Arabidopsis thaliana* seedlings. Arabidopsis seeds were disinfected with 70% (*v*/*v*) ethanol for 5 min, and then with 3% (*v*/*v*) chlorine bleach for 7 min, and they were finally washed five times with SDW. The assays were performed using 100 mm Petri dishes with two compartments. Arabidopsis seeds (three per replicate) were sown in the compartment of the Petri dishes that contained Murashige and Skoog medium supplemented with phytagel. After seed germination (≈72 h after being sown), 100 µL of a preculture of each *Streptomyces* strain, coming from a 4-day-old culture on ISP2 broth that was inoculated with 50 µL (≈2 × 10^6^ spores mL^−1^), were inoculated on a potato dextrose agar medium in the other compartment of the Petri dishes. Immediately, the Petri dishes were incubated at 28 ± 1 °C under a 16:8 h light/dark photoperiod for 10 d, under photosynthetically active radiation that was provided by LED bulbs at 1000 μmol m^−2^ s^−1^. The Petri dishes were placed at a 65° angle to avoid the obstruction of the aerial growth of the roots. The seedlings that were cultivated on Petri dishes without *Streptomyces* inoculum were used as control plants. The root length, the number of leaves, the shoot fresh weight, and the root fresh weight were evaluated in the seedlings 10 d after inoculation [35].

### 4.5. In Vitro and In Situ Growth Promotion of P. vulgaris by Streptomyces VOCs

The in vitro assays were performed as described above for *A. thaliana*, although the growth parameters (the root length, number of leaves, shoot fresh weight, root fresh weight, lateral root number, and lateral root density) were evaluated 5 days after incubation [35,55].

For the in situ assay, bean plants were grown in plastic pots (11 cm in diameter) that contained a sterilized mixture of loam soil, vermiculite, peat moss, and perlite (ratio: 5,1,1,1 *v*/*v*). The seedlings were maintained at 28 °C and under a 16:8 light/dark photoperiod, and they were watered at a 50% water-holding capacity. The seedlings emerged 5 d after being sown. Three days after emergence, 10 g of sterile oat flakes were placed on the substrate of each seedling and were immediately inoculated (10 mL) with the 4-day-old preculture of the *Streptomyces* strains described above. Each treatment consisted of five pots (replicates), with one seedling per replicate. The growth parameters (the root length, shoot length, stem and root fresh weights, hypocotyl diameter, and chlorophyll content as SPAD unit) were measured in the seedlings five weeks after inoculation [42,55].

### 4.6. In Vitro Growth Promotion of Arabidopsis thaliana Seedlings by Pure VOCs

The capacities of the pure 2-pentanone, 2-(methylthio) ethanol, *trans*-2-hexenal, 2,5-dimethylfuran, α-pinene, 2-methyl-3-pentanone, 3-methyl-2-butanol, geosmin + 2-methylisoborneol, valencene, and dimethyl disulfide at two concentration levels for the promotion of the growth of *A. thaliana* were evaluated in vitro. These VOCs were previously identified in the cultures of the tested *Streptomyces* strains by gas chromatography coupled to mass spectrometry in a previous study [13]. The experiment was performed in Petri dishes, as described above with bacteria, but by replacing the *Streptomyces* inoculum with filter papers that were impregnated with aqueous solutions of the pure VOCs. The concentration of the VOCs in the solutions was adjusted to 5 and 50 μL/L, respectively. The Petri dishes that contained seedlings and filter papers without VOCs were used as the control group. The plant-growth-promotion parameters were measured as indicated above.

### 4.7. Statistical Analysis

A completely randomized design was used with five treatments (*Streptomyces* strains and uninoculated control plants) and five replicates in both in vitro and in situ experiments. All of the measurements were made in triplicate. The results were subjected to an analysis of variance (ANOVA), and the means were separated by the Tukey test (*p* ≤ 0.05) by using Minitab 16 statistical software.

## 5. Conclusions

All of the *Streptomyces* strains produced IAA and siderophores and were able to solubilize phosphates, which presumably contributed to the plant development. The *Streptomyces* VOCs increased the growth parameters of the *A. thaliana* and *P. vulgaris* seedlings. The inoculation of the bean seedlings with the *Streptomyces* strains caused increases in the root and shoot lengths in the bean seedlings, which could be attributed to the synergistic effect of the VOCs and the nonvolatile compounds. Overall, the tested *Streptomyces* strains might be considered as potential biofertilizers. Further studies are needed to determine if the profiles of the VOCs of the *Streptomyces* strains are influenced by the plant species and by the synergistic effect of several pure VOCs.

## Figures and Tables

**Figure 1 plants-11-00875-f001:**
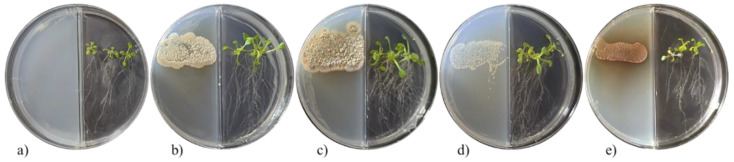
Appearance of *A*. *thaliana* seedlings after exposure for 10 days to VOCs emitted by *Streptomyces* strains. The assay was performed using divided Petri dishes; *A*. *thaliana* seeds were sown on the right, and *Streptomyces strains* were sown on the left of the dishes: (**a**) control; (**b**) *S*. *cangkringensis* CIAD–CA07; (**c**) *S*. *misionensis* CIAD–CA27; (**d**) *S. kanamyceticus* CIAD–CA45; and (**e**) *S. kanamyceticus* CIAD–CA48.

**Figure 2 plants-11-00875-f002:**
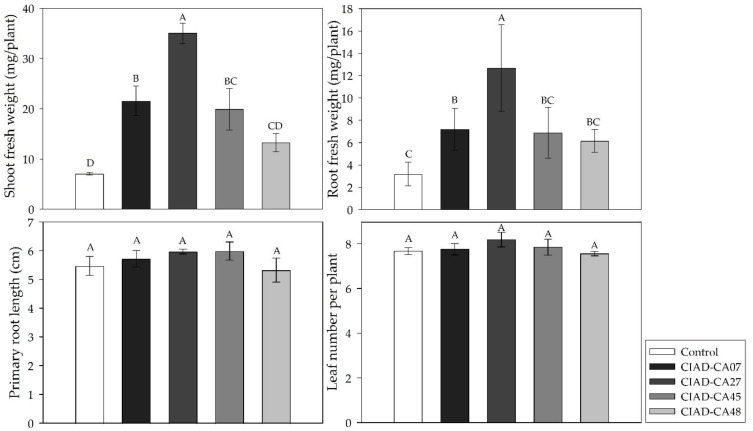
Effect of VOCs emitted by *Streptomyces* strains on biomass production in *Arabidopsis thaliana* seedlings. Mean values with the same letters are statistically similar. Thin upper bars indicate the standard deviations of the means.

**Figure 3 plants-11-00875-f003:**
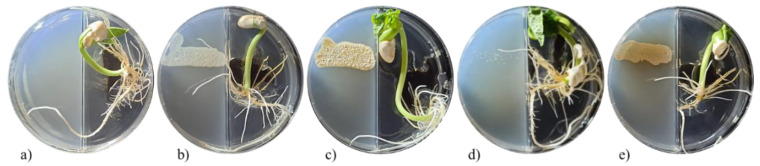
Appearance of *P*. *vulgaris* seedlings after exposure for 5 days to VOCs emitted by *Streptomyces* strains. The assay was performed using divided Petri dishes; *P*. *vulgaris* seeds were sown on the right, and *Streptomyces* strains on the left of the dishes: (**a**) control; (**b**) *S*. *cangkringensis* CIAD–CA07; (**c**) *S*. *misionensis* CIAD–CA27; (**d**) *S. kanamyceticus* CIAD–CA45; and (**e**) *S. kanamyceticus* CIAD–CA48.

**Figure 4 plants-11-00875-f004:**
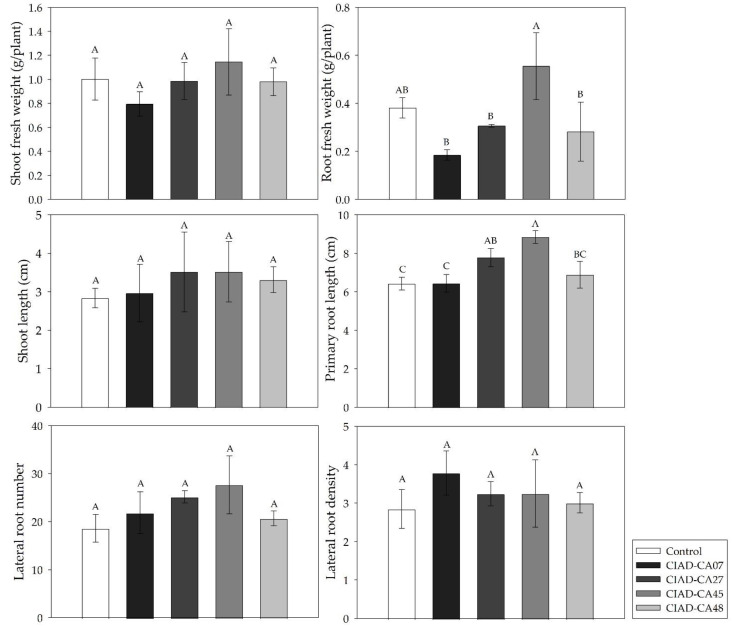
Effect of VOCs emitted by *Streptomyces* strains on biomass production in *P. vulgaris* seedlings. Mean values with the same letters are statistically similar. Thin upper bars indicate the standard deviations of the means.

**Figure 5 plants-11-00875-f005:**
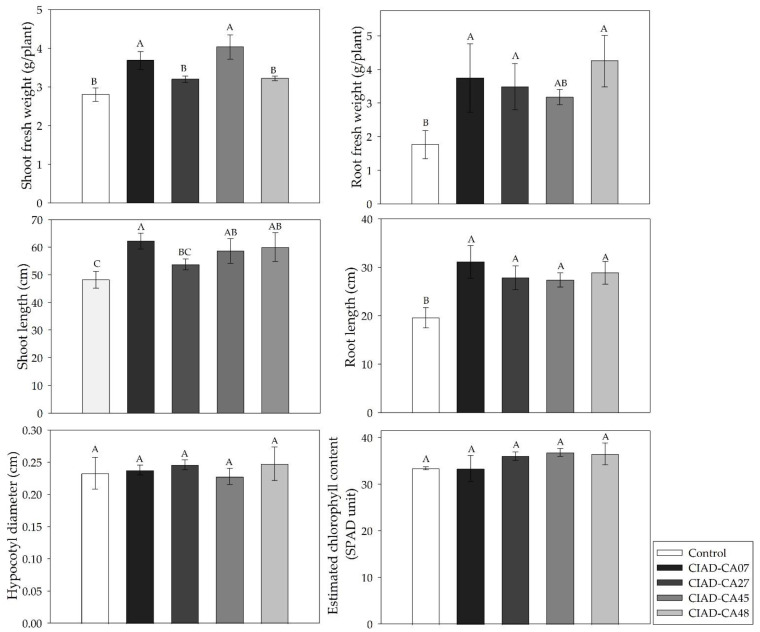
Effects of the inoculation of *P*. *vulgaris* seedlings with *Streptomyces* strains on biomass production. Mean values with the same letters are statistically similar. Thin upper bars indicate the standard deviations of the means.

**Figure 6 plants-11-00875-f006:**
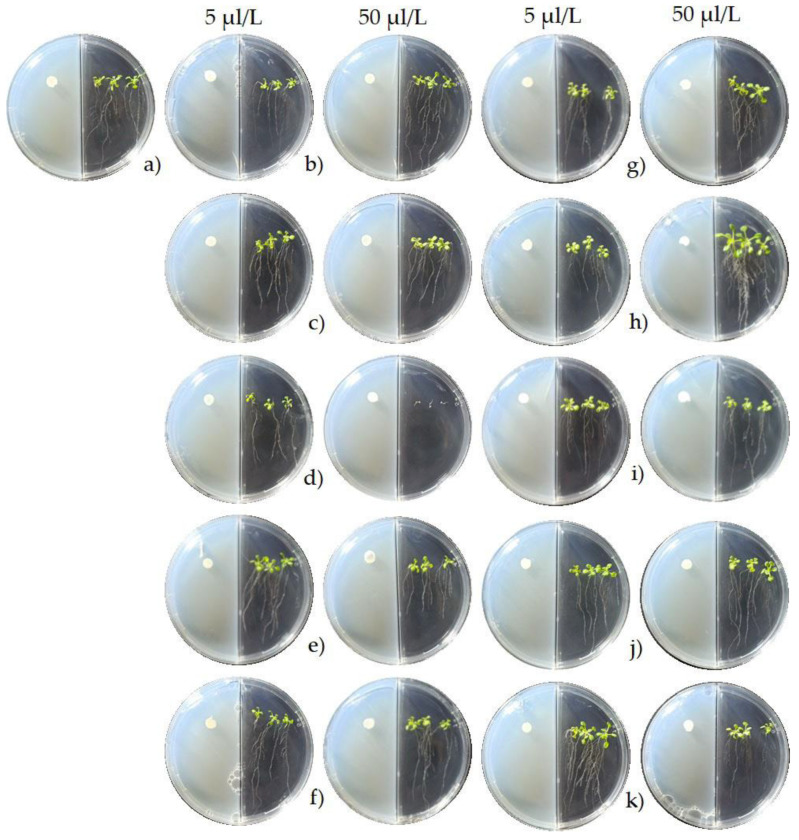
Appearance of *A*. *thaliana* seedlings after exposure for 10 days to pure VOCs (5 and 50 μL/L). The assay was performed using divided Petri dishes; *A*. *thaliana* seeds were sown on the right, and pure VOCs were sown on the left of the dishes: (**a**) control; (**b**) 2-pentanone; (**c**) 2-(methylthio) ethanol; (**d**) *trans*-2-hexenal; (**e**) 2,5- dimethylfuran; (**f**) α-pinene; (**g**) 2-methyl-3-pentanone; (**h**) 3-methyl-2-butanol; (**i**) geosmin + 2-methylisoborneol; (**j**) valencene; and (**k**) dimethyl disulfide.

**Figure 7 plants-11-00875-f007:**
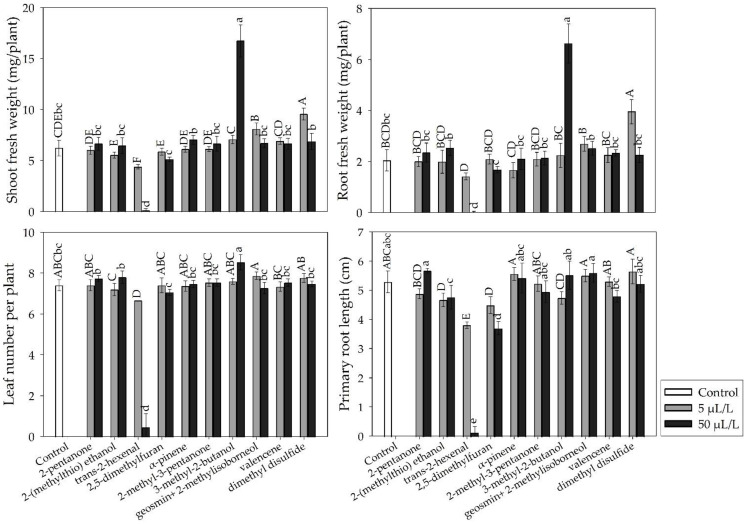
Effects of the exposure to pure VOCs for 10 days (5 and 50 μL/L concentrations) on biomass production in *Arabidopsis thaliana*. Mean values with the same letters are statistically similar. Thin upper bars indicate the standard deviations of the means.

**Table 1 plants-11-00875-t001:** Substances involved in plant growth promotion produced by *Streptomyces* strains.

Strains	Plant Growth Promotion Tests					
	Indole-3-Acetic Acid Production (mg/L) *	Siderophores			Nitrogen Fixing Activity	Phosphate Solubilization Activity
		Trihydroxamate	Dihydroxamate	Catechol		
*S. cangkringensis* CIAD-CA07	77.5 ± 15.0 a	+	−	−	−	+
*S. misionensis* CIAD-CA27	37.5 ± 5.0 b	+	−	−	−	+
*S. kanamyceticus* CIAD-CA45	17.5 ± 15.0 bc	+	−	−	−	+
*S. kanamyceticus* CIAD-CA48	10.0 ± 0.0 c	+	−	−	−	+

* mean value followed by standard error; values with the same letter within a column are not significant at *p* ≤ 0.05, according to Tukey test. −negative test; + positive test.

## Data Availability

The data presented in this study are available in the manuscript.

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
