# Peer review of "Growth Promotion of Phaseolus vulgaris and Arabidopsis thaliana Seedlings by Streptomycetes Volatile Compounds"

_plants, 2022, doi:10.3390/plants11070875_

Round 1

Reviewer 1 Report

Dear authors,

The manuscript entitled „Growth promotion of Phaseolus vulgaris and Arabidopsis thaliana seedlings by volatile compounds from anti-phytopathogenic Streptomycetes” Perez-Corral et al. describes the capacity of Streptomyces strains to produce substances involved in plant growth, solubilize phosphates, and fix nitrogen.

The idea of research is interesting and very good. Conclusions adequate to the conducted research. English language and style are fine.

Several changes are recommended, and some clarifications are required.

First, the authors must carefully read Instructions for authors.

Sometimes % is written with space or without space (line 22, line 24, 83 etc.).

Author Response

Manuscript Plants 1651126

Dear editor, thank you for sending us a copy of the revised draft of our manuscript titled “GROWTH PROMOTION OF Phaseolus vulgaris AND Arabidopsis thaliana SEEDLINGS BY STREPTOMYCETES VOLATILE COMPOUNDS” submit to the journal Plants. We appreciate the time and effort made by the panel of reviewers in reviewing the manuscript. As a result of their valuable observations/comments, the manuscript was greatly improved. The changes made to the manuscript in response to most of the suggestions provided by the reviewers are described below:

Response to Reviewer 1 Comments

Point 1: First, the authors must carefully read Instructions for authors.

Response 1: The manuscript has been adapted and supplemented in accordance with the guidelines requested by the journal Plants in the guide for authors, such as: Data availability statement: The following sentence was added “The data presented in this study are available upon request to the corresponding author”; Author contributions were changed using a letter designation instead of the full name for each author; fonts were checked.

Point 2: Sometimes % is written with space or without space (line 22, line 24, 83 etc.).

Response 2: The presentation of % in the manuscript was unified/homologated

Reviewer 2 Report

The article is interesting and fits in with the trends of sustainable agriculture in which the aim is to replace traditional fertilizers by microorganisms that show fertilizer activity and promote plant growth. The authors performed a number of experiments showing the positive effects of Streptomycetes and in particular VOCs emitted by these strains. The experimental layout is divided into two parts where Streptomycetes are used, followed by VOCs.

It is reasonably organised and written. However, my question is what was the main aim of the research: shoving the usefulness of the relatively new strains for stimulation of plant growth (?) or just checking how VOCs influence the plant growth (as stated in title). No anti-phytpoathogenic action was examined (probably this is the effect of the previous works of these authors). Therefore the title or article should be changed for example to: Growth Promotion of Phaseolus vulgaris and Arabidopsis thaliana Seedlings by Streptomycetes Volatile Compounds”

Line 20: Is IAA Indole-3-acetic acid (β-indolylacetic acid) ? Please insert abbreviation.

Acronyms/Abbreviations/Initialisms should be defined the first time they appear in each of three sections: the abstract; the main text; the first figure or table. When defined for the first time, the acronym/abbreviation/initialism should be added in parentheses after the written-out form.

Table 1 can be shortened as some information is already given in lines 66-70.

Line 78 please be careful when stating firmly that VOCs increased the biomass. The effect can be from other not detected compounds. Sometimes it is better to use wording such as "samples containing Streptomyces strains” . The same in lines 94, 98.

Line 100 and further - Add information about what is on the left and right sides of the petri dish in figs 1, 3 and 6.

Lines 112-113 write “ No differences were observed in……….” instead “  this and this were not positively influenced”

Materials and Methods: the methods are quite standard, the origin of the bacteria is well documented

Please give details concerning quality of the distilled water used in the experiments (lines 271-272) such as single/double distillation, UV-treatment, Filtration, other treatment. Please provide the name of apparatus used, producer name, city and country.

Lines 254-255 use correct font

Line 257 according to [48], The strains instead of These strains in lines 257 and 259

Line 277 plate assays using phosphate growth medium consisted of….

Line 285 Remove the full name from line 285, pay attention if U write indole acetic acid and indole—3-acetic acid – unify the name

Line 287 using methods described previously in [30] and [44] (optionally [30,44]

Line 290 4mL of ISP Medium 2 instead of International Streptomyces……

Line 294-295 replace [  ] with (  )

Lines 298-300 Please add information of what wavelength was used as a zero in baseline

Other remarks: Formatting and style

Lines 319 and 333 day/night rather than L:D

Line 320 : check fonts

Please pay attention to the fonts U are using.

Fonts in the title:

Please pay attention to fonts, check already printed works.

Correct fonts in lines 26, 122, 248, 254-255, 320, 350-351, 367

Line 26 : 3-Methyl-2-butanol please - check font

Line 376 Author contributions; please use a letter designation instead of the full name for each author. Example as below:

Conceptualization, T.T. and H.K.; methodology, T.T., L.Y. and H.S.; investigation,

T.T., L.Y. and H.S.; writing—original draft preparation, T.T.; writing—review and editing, T.T.

and H.K.; visualization, T.T. and L.Y.; supervision, T.T. and H.K.; funding acquisition, T.T and H.K.

All authors have read and agreed to the published version of the manuscript.

Minor English corrections are required.

Author Response

Manuscript Plants 1651126

Dear editor, thank you for sending us a copy of the revised draft of our manuscript titled “GROWTH PROMOTION OF Phaseolus vulgaris AND Arabidopsis thaliana SEEDLINGS BY STREPTOMYCETES VOLATILE COMPOUNDS” submit to the journal Plants. We appreciate the time and effort made by the panel of reviewers in reviewing the manuscript. As a result of their valuable observations/comments, the manuscript was greatly improved. The changes made to the manuscript in response to most of the suggestions provided by the reviewers are described below:

Response to Reviewer 2 Comments

Point 1: What was the main aim of the research?

Response 1: The study aimed was to evaluate the ability of VOCs emitted by four Streptomyces strains to produce to promote plant growth of Arabidopsis thaliana and Phaseolus vulgaris seedlings, as well as to trying to elucidate which of the VOCs emitted by the strains might be involved in promoting plant growth.

Point 2: The title or article should be changed?

Response 2: The title of the manuscript was rewritten based on the new suggested title, remaining as follows “GROWTH PROMOTION OF Phaseolus vulgaris AND Arabidopsis thaliana SEEDLINGS BY STREPTOMYCETES VOLATILE COMPOUNDS”. Since a section on antifungal tests was not addressed in this study; we just wanted to highlight that our Streptomyces strains also have antagonistic capacity against phytopathogenic fungi previously reported in studies conducted by our working group.

Point 3: Is IAA Indole-3-acetic acid (β-indolylacetic acid)?

Response 3: In response to this apt observation, the full name was written before the abbreviation.

Point 4: Table 1 can be shortened as some information is already given in lines 66-70.

Response 4: Based on this apt observation/suggestion we decided to leave Table (1) since the information for comparative purposes is more readable form; therefore, we opted to modify the text presented in lines 66-73, being as follows " Results of nitrogen-fixing and phosphate solubilization by Streptomyces strains, as well as, detection of catechol and hydroxamate type siderophore, are shown in Table 1”.

Point 5: Line 78 please be careful when stating firmly that VOCs increased the biomass. The effect can be from other not detected compounds. Sometimes it is better to use wording such as "samples containing Streptomyces strains”. The same in lines 94, 98.

Response 5: Since the Streptomyces - seedling interaction tests, were conducted in Petri dishes divided in half, that is, there was a physical barrier; therefore, the only contact was through the VOCs in the headspace, so we assume that the promotion was due to the volatiles emitted by Streptomyces.

Point 6: Line 100 and further - Add information about what is on the left and right sides of the petri dish in figs 1, 3 and 6.

Response 6: The captions of the figures (1, 3 and 6) were supplemented:

Figure 1, with the following description “The assay was performed using divided Petri dishes, A. thaliana seeds were sown on the right and Streptomyces strains on the left of the plates”.

Figure 3, with the following description “The assay was performed using divided Petri dishes, P. vulgaris seeds were sown on the right and Streptomyces strains on the left of the dishes”.

Figure 6, with the following description “The assay was performed using divided Petri dishes, A. thaliana seeds were sown on the right and pure VOCs on the left of the dishes”.

Point 7: Lines 112-113 write “No differences were observed in……….” instead “ this and this were not positively influenced”

Response 7: The content of the lines 117-118 was modified, remaining as follows “No differences were observed in hypocotyl diameter and chlorophyll content in bean seedlings”.

Point 8: Please give details concerning quality of the distilled water used in the experiments (lines 271-272) such as single/double distillation, UV-treatment, Filtration, other treatment. Please provide the name of apparatus used, producer name, city and country.

Response 8: The required information was added incorporated into the manuscript as follows “(SDW – Osmosis and UV treatment (Ultrapure Water Production System (FES-TA Model HGL-UP-TOC-100, Chihuahua, Mexico))”.

Point 9: Line 257 according to [48], The strains instead of These strains in lines 257 and 259

Response 9: This suggestion was accepted

Point 10: Line 277 plate assays using phosphate growth medium consisted of….

Response 10: The suggestions was accepted

Point 11: Line 285 Remove the full name from line 285, pay attention if U write indole acetic acid and indole—3-acetic acid – unify the name

Response 11: This suggestion was accepted and was unified as “Indole-3-acetic acid”

Point 12: Line 287 using methods described previously in [30] and [44] (optionally [30,44]

Response 12: This suggestion was accepted

Point 13: Line 290 4mL of ISP Medium 2 instead of International Streptomyces……

Response 13: This suggestion was accepted

Point 14: Line 294-295 replace [  ] with (  )

Response 14: This suggestion was accepted

Point 15: Lines 298-300 Please add information of what wavelength was used as a zero in baseline

Response 15: The wavelength at baseline was the same “535 nm”.

Point 16: Lines 319 and 333 day/night rather than L:D

Response 16: L:D was replaced by “light/dark” in the manuscript

Point 17: Please pay attention to fonts, check already printed works.

Response 17: Fonts in the manuscript were checked.

Point 18: Line 376 Author contributions; please use a letter designation instead of the full name for each author

Response 18: Author contributions were changed using a letter designation instead of the full name for each author.
